# Patient satisfaction with the quality of care received is associated with adherence to antidepressant medications

**Macarius M. Donneyong**[1]*, **Mary Bynum**[2], **Ameena Kemavor**[3], **Norah L. Crossnohere**[4], **Anne Schuster**[5], **John Bridges**[5]

1 Division of Outcomes and Translational Sciences, College of Pharmacy, The Ohio State University, Columbus, OH, United States of America, 2 Healthcare Management, Franklin University, Columbus, Ohio, United States of America, 3 ADAMH Board of Franklin County, Columbus, OH, United States of America, 4 Division of General Internal Medicine, The Ohio State College of Medicine, Columbus, Ohio, United States of America, 5 Department of Biomedical Informatics, The Ohio State College of Medicine, Columbus, Ohio, United States of America

* donmacarius@yahoo.co.uk, donneyong.1@osu.edu

**Data Availability Statement:** All the data analyzed for this study can be publicly accessed at the MEPS website: https://meps.ahrq.gov/mepsweb/data_stats/download_data_files_results.jsp?

## Abstract

### Background

There is a paucity of evidence on the association between satisfaction with quality of care and adherence to antidepressants.

### Objectives

To examine the association between patient satisfaction with healthcare and adherence to antidepressants.

### Methods

A cohort study design was used to identify antidepressant users from the 2010-2016 Medical Expenditure Panel Survey data, a national longitudinal complex survey study design on the cost and healthcare utilization of the noninstitutionalized population in the United States. The Consumer Assessment of Healthcare Providers and Systems were used to measure participants' satisfaction with access and quality of care, patient-provider communication and shared decision-making (SDM). Patients were considered satisfied if they ranked the quality of care at $\geq 9$ (range: 0[worst]– 10[best]). Antidepressant adherence was measured based on medication refill and complete discontinuation. MEPS sampling survey-weighted multivariable-adjusted logistic regression models were used to calculate the odds ratios (ORs) and 95% confidence intervals (CIs) for the associations between satisfaction and adherence to antidepressants. We tested for the potential presence of reverse associations by restricting the analysis to new users of antidepressants. The roles of patient-provider communication and SDM on the satisfaction-adherence association were examined through structural equation models (SEM).

cboDataYear=All&cboDataTypeY=1%2CHousehold
+Full+Year+File&buttonYearandDataType=
Search&cboPufNumber=All.

**Funding:** The author(s) received no specific funding for this work.

**Competing interests:** The authors have declared that no competing interests exist.

## Results

Among 4,990 (weighted counts = 8,661,953) antidepressant users, 36% were adherent while 39% discontinued antidepressants therapy. Half of antidepressant users were satisfied with the healthcare received. Satisfied patients were 26% (OR = 1.26, 95%CI: 1.08, 1.47) more likely to adhere and 17% (OR = 0.83, 95%CI: 0.71, 0.96) less likely to discontinue, compared to unsatisfied antidepressant users. Patient satisfaction was also associated with higher odds (OR = 1.41, 95%CI: 1.06, 1.88) of adherence among a subgroup of new users of antidepressants. The SEM analysis revealed that satisfaction was a manifestation of patient-provider communication *(β = 2.03, P-value<0.001)* and SDM *(β = 1.14, P-value<0.001)*.

## Conclusions

Patient satisfaction is a potential predictor of antidepressant adherence. If our findings are confirmed through intervention studies, improving patient-provider communication and SDM could likely drive both patient satisfaction and adherence to antidepressants.

## Introduction

Depression represents one of the greatest sources of morbidity in the US and worldwide [1, 2]. Nearly 12% of older adults–over 6 million in the US–have depression, a condition which confers significant increases in both morbidity and mortality [3, 4]. A combination of psychotherapy and antidepressant medications is the mainstay of treatment for major depressive disorder (MDD). However, nearly a third of MDD patients treated with antidepressants do not achieve remission [5]. This could be explained partly by the fact that only 21–31% of patients maintain adherence to antidepressant therapy within 6–12 months after treatment [6]. Suboptimal adherence to antidepressants is associated with a 2 to 8-fold higher risk of relapse/recurrence and 14–20% higher rates of all-cause hospitalizations or emergency room visits [7]. Thus, suboptimal adherence to antidepressants is a significant public health problem that needs to be addressed in order to treat depression and prevent its adverse impact of disability, morbidity, and mortality.

Providers, including those who provide mental health care, are increasingly required to conduct patient satisfaction surveys as part of patient-centered care models [8–10]. Patient satisfaction data is typically used by payers for reimbursement purposes while patients use this information to make decisions on which provider to choose [8–11]. Thus, patient satisfaction ratings data could be rich data sources for examining the role of satisfaction in patients' adherence to antidepressant therapy [12, 13]. Knowledge from this type of research could unlock the utility of patient satisfaction data beyond meeting mandatory reimbursement requirements by providing insights on the potential effect of patient satisfaction on medication adherence.

Medication adherence functions as a shared agreement between the patient and their clinician [14, 15]. Patient-provider communication and shared decision-making (SDM) are the core tenets of this patient-provider relationship [16]. Patient-provider communication and patient involvement in SDM are also thought of as the key drivers of patient satisfaction with healthcare in the general population [17, 18]. Both patient-provider relationship constructs have been shown to be associated with adherence to ADs [19]. However, there are no published studies to elucidate the potential mechanisms through which patient satisfaction ratings

might influence patients' adherence to AD therapy. There are also no published studies that have examined the role of patient-provider communication and SDM in the association between satisfaction and antidepressant adherence. To address the aforementioned research gaps, we sought to: 1) assess whether patient satisfaction ratings are associated with antidepressant adherence; and 2) examine the role of patient-provider communication and SDM in the satisfaction-adherence association.

## Methods

### Conceptual framework

We used a Directed Acyclic Graph (DAG) to identify and illustrate the inter-relationships between individual-, provider- and healthcare system-level determinants of medication adherence and how these multilevel factors operate to drive adherence to antidepressants directly or indirectly via their effects on patient satisfaction with healthcare quality (**Fig 1**).

In this DAG, we theorize that patient satisfaction with healthcare quality is a manifestation of patients' interactions with providers and the healthcare system within which they receive healthcare. Those patients who experience positive patient-provider interactions and feel supported by their healthcare systems would be more likely to be satisfied with healthcare than those who do not have positive interactions with healthcare systems [20]. The more satisfied patients are, the more likely they would be to adhere to antidepressant therapy and develop positive perceptions of antidepressant therapy. Alternatively, nonadherent patients would be less likely to benefit from antidepressant therapy and thus may form negative perceptions about antidepressant therapy which could in turn influence how they rate the quality of healthcare. In other words, patients' ratings could be influenced by their perception of the quality of care and benefits of antidepressant therapy and thus creating a feedback loop in the satisfaction-AD adherence relationship. The analysis described below was informed by the relationships as described in this DAG (**Fig 1**).

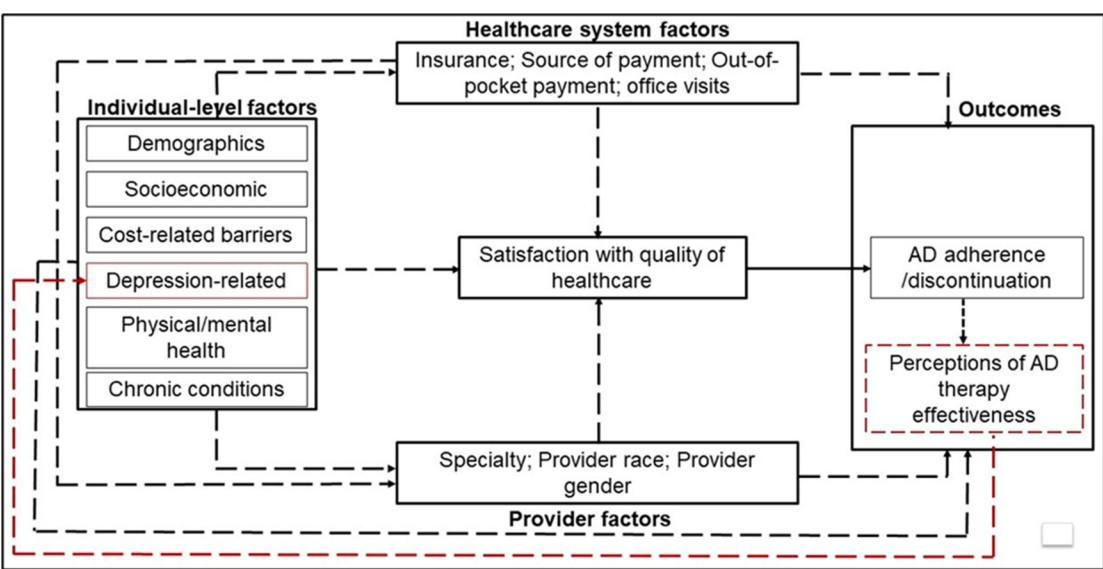

**Fig 1. Directed acyclic graph (DAG).** We theorize that patient satisfaction with healthcare quality is a manifestation of patients' interactions with providers and the healthcare system within which they receive healthcare. The direct path from patient satisfaction to antidepressant adherence/discontinuation was quantified, all other relationships (dashed arrows) were accounted for in the analysis as covariates. Potential reverse association is represented as a feedback loop from patient perceptions of the effectiveness of antidepressant to depression-related factors (red dashed line).

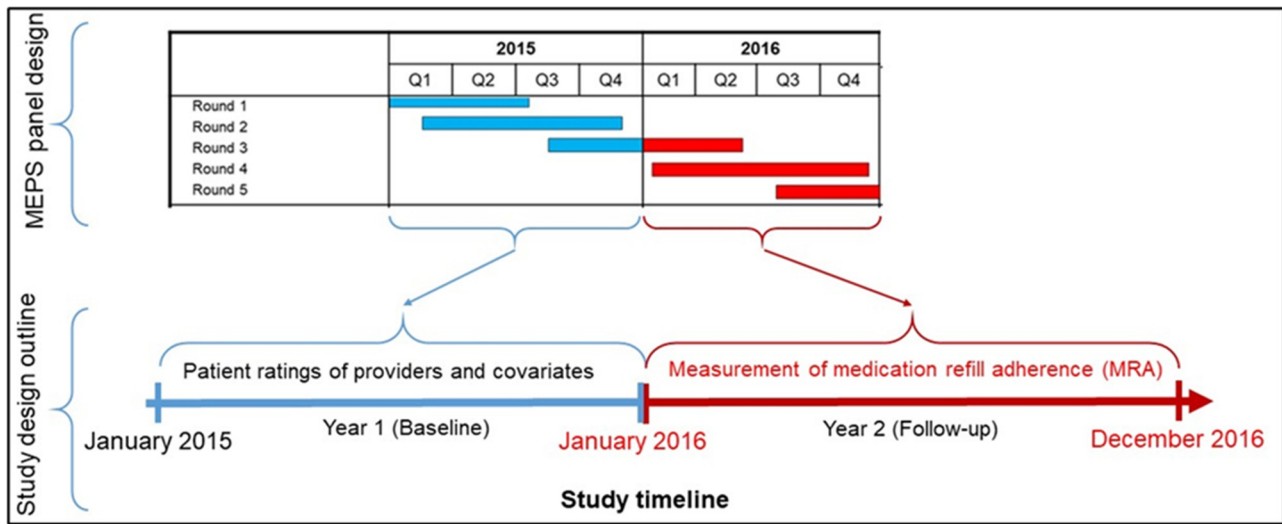

**Fig 2. Study design.** In this illustration, data collected from rounds 1–3 in 2015 are used to define patient ratings of providers, patient-provider relationships, access to care, and all covariates. The 2016 survey data are used to define medication refill adherence (MRA) based on the days' supply of filled drugs.

## Data source, setting

We analyzed data from the Household Component of the Medical Expenditure Panel Survey (MEPS-HC) from 2010 to 2017 [21]. The MEPS-HC data is collected from a nationally representative sample of households through an overlapping complex panel survey design (**Fig 2**).

MEPS used the computer-assisted personal interview (CAPI) and self-administered supplemental paper questionnaires (SAQ) respectively to conduct five rounds of interviews during a two-year period. Published response rates of the surveys ranged from 48.3 to 58.6.0% (mean 55.2 ± 4.6%) during the time period analyzed [22]. These surveys collect information on self-reported health status, medical conditions, health insurance status, healthcare access, prescription medication use, access to and satisfaction with providers.

MEPS adapted the healthcare quality measures from the health plan version of Consumer Assessment of Healthcare Providers and Systems (CAHPS®) survey to collect participants' perspectives about access to care and the quality of healthcare they received from doctors and other healthcare providers. CAHPS is a reliable and valid tool for capturing information about health plans' performances from racially/ethnically diverse consumers [23–26]. We excluded participants who did not visit a doctor's office or clinic in year 1 since they were ineligible to rate the quality of healthcare received from providers.

## Study sample

Among eligible participants, we included those who had a self-reported diagnosis of Major Depressive Disorder (MDD). All medical conditions, including MDD are captured in the medical conditions file of MEPS and classified using the *International Statistical Classification of Diseases*, *Tenth Revision*, *Clinical Modification [ICD-10-CM] codes* [27]. We identified MDD based on the following ICD-10-CM codes: F32, F33, F34, and F39.

## Study design

Every year, MEPS selects a new sample or panel of households that are followed up for two full calendar years. As such, six consecutive two-year panels were established during the 2010–

2017 study period that was chosen for our analysis. During this two-year timeframe, each household respondent provided information on behalf of all household members through five in-person interviews, also known as survey rounds. **Fig 2** is an illustration of how the MEPS panel design data was leveraged to create a cohort of antidepressant users. In this illustration, two years of data (e.g., 2015–2016) from five rounds of surveys are combined to create a cohort of participants who filled at least one prescription of antihypertensive medication. The first calendar year for each participant included in our analysis was defined as the baseline period for measuring covariates and primary exposures whereas the second year was set as the follow-up period for the outcome. For example, in **Fig 2**, data collected from rounds 1–3 in 2015 (Year 1) were used to measure all potential confounders as well as patient satisfaction with the healthcare that they had received (primary exposure). The 2016 data (Year 2) were then used to define adherence measures (outcome) based on medication refills and the days' supply of filled antidepressant medications.

## Assessment of antidepressant medication use

MEPS collected information about prescription medication use from participants during each survey cycle. To ascertain the veracity of self-reported medication use, MEPS data collectors obtained consent from participants to contact the pharmacies at which these medications were filled. Upon receiving patient consent, MEPS staff collected the following information from the pharmacies that filled the prescriptions for participants: payments, payers, date each prescription was filled, quantity dispensed, the National Drug Code (NDC) [28]. The pharmacies also provided information on the number of times medications were filled within a given calendar year, as such MEPS captured all the medications that were refilled within a given calendar year [28].

To identify antidepressant users from the MEPS dataset, we classified all patients who had records of filling at least one antidepressant agent during both their baseline and follow-up periods. Overall antidepressant use was defined using the following therapeutic subclass codes: 76, 208, 209, 249, 306, 307, 308. Further, we applied the Canadian Network for Mood and Anxiety Treatments (CANMAT) guidelines for pharmacologic treatment of depression to classify antidepressant users into first-line versus second/third-line antidepressant therapy. Participants who used an antidepressant from any of the following sub-therapeutic classes were classified as users of first-line antidepressant therapy: agomelatine, bupropion, citalopram, desvenlafaxine, duloxetine, escitalopram, fluoxetine, fluvoxamine, mianserin, milnacipran, mirtazapine, paroxetine, sertraline, venlafaxine, or vortioxetine [29].

## Outcomes

Medication adherence is a dynamic process by which patients take their medications as prescribed and is comprised of three phases: the initiation (obtaining the medication and taking the first dose); implementation (taking each prescribed dose in a timely manner); and discontinuation (ceasing to take the medication) phases [30]. Because our analysis involved antidepressant users, we focused on the implementation and discontinuation phases of the medication adherence process.

a. *Implementation of* antidepressant *therapy*: this was defined as antidepressant medication refill adherence (MRA). Detailed information on the type, dosage, and payment for each filled prescription during follow-up were used to calculate MRA was as the percent of total days' supply divided by number of days of study participation (365 days) [31, 32]. Thus, measures of MRA in this study reflect long-term adherence to antidepressants during a

365-day observation period. MRA has been previously used to measure adherence from the MEPS dataset [33, 34]. An overall MRA was obtained as the average of MRAs calculated for separate therapeutic classes of antidepressant medications if a participant was using more than one antidepressant agent from multiple therapeutic classes. Patients were considered to be adherent to antidepressant medications if their overall MRA was $\geq$80% [35–37].

b. *Antidepressant discontinuation*: Participants who reported using antidepressants in year 1, but not in year 2, were considered to have completely discontinued antidepressant use in year 2.

## Measurement of satisfaction

At the midpoint of study years, patients were asked to respond CAHPS questionnaire. CAHPS survey assesses patient satisfaction across different dimensions, ranging from physician communication to the quality of health plan customer service [38]. MEPS participants who reported having a usual source of care (USC) were asked to rate the healthcare that they had received from all doctors and other health providers, from 0 (worst health care possible) to 10 (best health care possible). We used the top-box approach to dichotomize the scores for this global question into satisfied ($\geq$9) versus unsatisfied (<9) participants who reported using antidepressants–this classification approach is consistent with the Centers of Medicaid and Medicare Services' (CMS') categorization of survey responses to the CAHPS questionnaire [39, 40].

## Measurement of patient-provider relationships and covariates

Communication between patients and providers and the engagement of patients in SDM are core tenets of the patient provide-relationship. CHAPS surveys and other questionnaire on access to health were used to elicit patients' assessment of their communication and involvement in SDM. We applied previously published methods [41] to define constructs of communication [39] and SDM [41] based on patient responses to the CHAPS and access to health questionnaires [41]. Based on our proposed conceptual framework, several variables were identified as potential covariates, these included several of the variables listed in the World Health Organization's (WHO's) multidimensional framework of adherence [42]. Given that medication adherence is a complex interplay of multilevel factors, we grouped the measured covariates at the individual patient, healthcare provider and healthcare system levels. We defined potential covariates (Table 1) from the data collected during baseline (year 1) to precede the period during which adherence was assessed. Poverty was defined as total family income less than 200% of the federal poverty level. Participants were considered to be of poor physical health if their response to a question to self-rate their physical health at the time of survey was "fair or poor health". Participants were also asked to self-rate their mental health at the time of survey, those who indicated that their mental health was "fair" or "poor", were considered to be of poor mental health. The rest of the covariates were measured using standard MEPS definitions.

## Missing data

Between 8% and 12% of participants were missing responses to questions about rating healthcare, patient-clinician communication and shared-decision making. We imputed missing values via random selection methods [43, 44] to conserve the study sample size with the assumption that values were missing at random. The distributions of scores for healthcare

**Table 1. Distribution of baseline covariates by levels of satisfaction with quality of healthcare among users of antidepressants in the medical expenditure panel survey.**

| | Overall | Unsatisfied (n = 2494) | | Satisfied (n = 2487) | | SDT |
|---|---|---|---|---|---|---|
| | | n* | %+ | n* | %+ | |
| **Individual-level factors** | | | | | | |
| *Demographic factors* | | | | | | |
| Age categories | | | | | | |
| 18–44 | 1420 | 841 | 33 | 579 | 19 | 0.32 |
| 45–64 | 2237 | 1108 | 46 | 1129 | 45 | 0.02 |
| 65+ | 1333 | 514 | 21 | 819 | 35 | 0.34 |
| Female | 3537 | 1707 | 68 | 1830 | 71 | 0.07 |
| Race/ethnicity | | | | | | |
| White | 3445 | 1709 | 69 | 1736 | 70 | 0.01 |
| Black | 628 | 323 | 13 | 305 | 12 | 0.02 |
| Hispanic | 701 | 350 | 14 | 351 | 14 | 0.00 |
| Other Race | 207 | 112 | 4 | 95 | 4 | 0.00 |
| Geographic region of residence | | | | | | |
| Midwest | 1322 | 651 | 13 | 671 | 15 | 0.06 |
| Northeast | 700 | 327 | 29 | 373 | 28 | 0.01 |
| South | 1894 | 920 | 37 | 974 | 38 | 0.01 |
| West | 1074 | 565 | 22 | 509 | 20 | 0.05 |
| Married | 2491 | 1182 | 48 | 1309 | 52 | 0.05 |
| Language—non-English speaking | 235 | 92 | 54 | 143 | 59 | 0.11 |
| *Socioeconomic and cost-related factors* | | | | | | |
| Educational level—college and above | 2560 | 1282 | 60 | 1278 | 59 | 0.01 |
| Unemployed | 3086 | 1490 | 17 | 1596 | 13 | 0.13 |
| Poverty (< 200% of the federal poverty level) | 1105 | 591 | 55 | 514 | 60 | 0.1 |
| Delay in purchasing prescribed medicine due to cost | 279 | 175 | 10 | 104 | 7 | 0.13 |
| Delay in seeking medical care due to cost | 397 | 239 | 6 | 158 | 3 | 0.13 |
| *Antidepressant-related* | | | | | | |
| Prior antidepressant use | 3050 | 1459 | 60 | 1591 | 63 | 0.08 |
| Used first-line antidepressant± | 2056 | 966 | 28 | 1090 | 32 | 0.03 |
| *Mental health-related* | | | | | | |
| Poor physical health | 1050 | 594 | 20 | 456 | 14 | 0.17 |
| Poor mental health | 651 | 384 | 14 | 267 | 8 | 0.2 |
| Cognitive limitations | 938 | 516 | 18 | 422 | 13 | 0.13 |
| Depression | 2392 | 1216 | 56 | 1176 | 50 | 0.12 |
| Anxiety | 315 | 145 | 7 | 170 | 8 | 0.02 |
| Psychotherapy | 954 | 540 | 42 | 414 | 40 | 0.04 |
| PHQ2 score high ($\geq$3) | 1257 | 761 | 27 | 496 | 16 | 0.27 |
| *Chronic comorbidities* | | | | | | |
| Hypertension | 2687 | 1300 | 49 | 1387 | 53 | 0.10 |
| Coronary heart disease | 485 | 225 | 7 | 260 | 10 | 0.09 |
| Angina | 260 | 132 | 5 | 128 | 5 | 0.02 |
| Myocardial infarction | 353 | 166 | 6 | 187 | 7 | 0.07 |
| Other heart disease | 949 | 454 | 19 | 495 | 19 | 0.02 |
| Stroke | 482 | 244 | 8 | 238 | 9 | 0.02 |
| Diabetes | 1067 | 490 | 19 | 577 | 20 | 0.03 |
| Arthritis | 2646 | 1276 | 50 | 1370 | 54 | 0.08 |

*(Continued)*

**Table 1.** (Continued)

| | Overall | Unsatisfied (n = 2494) | | Satisfied (n = 2487) | | SDT |
|---|---|---|---|---|---|---|
| | | n* | %+ | n* | %+ | |
| Asthma | 927 | 475 | 19 | 452 | 16 | 0.06 |
| Chronic bronchitis | 371 | 183 | 7 | 188 | 7 | 0.01 |
| Cancer | 852 | 387 | 17 | 465 | 22 | 0.12 |
| **Provider-level factors** | | | | | | |
| Provider specialty characteristics | | | | | | |
| General medical doctor | 1584 | 735 | 35 | 849 | 37 | 0.05 |
| Psychiatrist/Mental health specialist | 449 | 188 | 9 | 261 | 11 | 0.08 |
| Other medical doctor | 130 | 68 | 3 | 62 | 3 | 0.06 |
| Other clinician | 126 | 70 | 4 | 56 | 3 | 0.05 |
| Unknown (facility or person-in-facility) | 2338 | 1172 | 51 | 1166 | 48 | 0.07 |
| Provider-patient gender concordance | 1861 | 922 | 39 | 939 | 43 | 0.06 |
| Provider-patient race/ethnicity concordance | 1657 | 771 | 43 | 886 | 41 | 0.05 |
| **Healthcare system factors** | | | | | | |
| Uninsured | 351 | 210 | 7 | 141 | 4 | 0.12 |
| Have usual source of payment | 4581 | 2219 | 93 | 2362 | 96 | 0.13 |
| Medicaid | 970 | 516 | 58 | 454 | 63 | 0.10 |
| Medicare | 1020 | 458 | 15 | 562 | 11 | 0.09 |
| Private insurance | 2486 | 1160 | 18 | 1326 | 21 | 0.07 |
| Average annual out-of-pocket payment | 4990 | 499.9 | 557 | 476.4 | 511 | 0.03 |
| Average annual number of hospital visits | 4990 | 12.7 | 13 | 12.2 | 12 | 0.05 |

Abbreviations: SDT, standardized difference test; PHQ, Patient Health Questionnaire

*Represent mean for continuous variables—average annual out-of-pocket payment and average annual number of hospital visits

+Represent standard deviation for continuous variables—average annual out-of-pocket payment and average annual number of hospital visits

± Use of first-line antidepressant was determined from the first antidepressant filled in year 2 since antidepressant adherence was measured in year 2.

ratings, communication and shared-decision making did not change after random selection imputation.

## Statistical analysis

First, we described the study population by the distribution of baseline covariates, applied regression methods to assess the associations between patient satisfaction and antidepressant adherence and examined the potential mechanisms that explain the satisfaction-adherence relationships. We described the distribution of patient satisfaction scores by MRA levels, overall and by race/ethnicity. Standardized difference tests (SDTs) were used to assess the balance of baseline covariates between satisfied and unsatisfied patients.

Second, we used multivariable-adjusted logistic regression models, weighted by MEPS survey weights [45], to measure the odds ratios (ORs) and 95% confidence intervals (CIs) for the associations between patient satisfaction (as a binary predictor) and adherence to ADs. We repeated this analysis by modeling complete antidepressant discontinuation as the outcome. In sensitivity analysis, we tested for the potential presence of reverse associations which could arise when patients rate the quality of healthcare based on their perceptions of the effectiveness of antidepressant therapy. The potential presence of reverse association between adherence and patient-clinician relationships is illustrated in the DAG (Fig 1) by a feedback loop from perceptions of antidepressant therapy, developed after the use of antidepressants in year 2,

back to patients' ratings of the quality of healthcare. For example, patients who perceive antidepressant therapy to be effective after antidepressant use would be more likely to adhere and to feel more satisfied with their healthcare than if they had developed negative perceptions of antidepressant therapy. To test for the presence of a reverse association we repeated the analysis among new users of antihypertensive medications. We assumed that the reported satisfaction levels reported in year 1, prior to antidepressant use in year 2, were not influenced by patients' perceptions of antidepressant effectiveness since new users did not use antidepressants in year 1. Thus, any observed associations between patient satisfaction and antidepressant adherence among new users could not have been influenced by the patients' perceptions of antidepressant therapy.

We applied structural equation modeling (SEM) to examine the role of patient-provider communication and SDM on the hypothesized association between satisfaction and antidepressant adherence. We hypothesized that patient satisfaction is a manifestation of modifiable patient-provider interactions (communication and patient involvement in SDM process) and that these modifiable patient-provider interactions drive the associations between satisfaction and antidepressant adherence through direct and indirect pathways. We quantified the magnitude of the direct, indirect, and total effects of patient-provider constructs and satisfaction on antidepressant adherence independent of baseline covariates through a multivariable-adjusted SEM. We tested the fitness of each hypothesized model to the data based on the Root Mean Square Error of Approximation (RMSEA), Tucker-Lewis Index (TLI) and the Comparative fit index (CFI). We implemented the SEM analysis using AMOS v26.0 (IBM).

## Results

The final analysis included data from 4,990 (weighted count = 8,661,953) users of AD; 1,931 (weighted count = 3,403,982) of these discontinued using antidepressants in year 2. Conversely, 1,940 (weighted count = 3,328,486) participants started using antidepressants in year 2 after not previously reporting using antidepressants in year 1 (i.e. new users of ADs). Half (50%) of the antidepressant users were satisfied with the healthcare they had received from providers.

The distribution of baseline covariates is represented in Table 1. The standardized difference tests showed that the majority of the baseline covariates were balanced (based on a threshold of standardized difference >0.10) between satisfied and unsatisfied antidepressant users except for some demographic, access to care barriers, depression-related and comorbidities. Satisfied patients were less likely to have experienced cost-related delays in getting prescribed medications or receiving medical treatment, less likely to be uninsured, less likely to have poor mental health, less likely to have cognitive limitations and less likely to have severe depression symptoms. On the other hand, satisfied patients also tended to be older, more likely to be non-English speakers and more likely to have had a usual source of payment for healthcare services. The distributions of the rest the baseline covariates were similar between satisfied and unsatisfied antidepressant users. Notably, the distributions of provider specialty, gender and race concordance between patients and providers were similar between satisfied and unsatisfied patients. Similarly, all healthcare system factors (except insurance status and usual source of payment) were nearly equally distributed between satisfied and unsatisfied antidepressant users.

### Associations between patient satisfaction with healthcare and adherence to antidepressants

Only 36% of antidepressant users were adherent–the prevalence of antidepressant adherence was higher among those who were satisfied (46%) compared to those not satisfied (38%) with

**Table 2. Associations between patient satisfaction with healthcare and adherence to antidepressant (AD) medications.**

| AD users | Adherence to ADs | | | Discontinuation of ADs | | |
|---|---|---|---|---|---|---|
| | Prevalence of adherence by patient satisfaction level, % | | OR (95% CI)* | Prevalence of discontinuation by patient satisfaction level, % | | OR (95% CI)* |
| | Satisfied | Unsatisfied | | Satisfied | Unsatisfied | |
| All users, n = 4990 | 44.8 | 40.3 | 1.26 (1.08, 1.47) | 36.0 | 41.5 | 0.83 (0.71, 0.96) |
| New users, n = 1940 | 26.2 | 21.1 | 1.41 (1.06, 1.88) | n/a | n/a | n/a |

Abbreviations: CI, confidence interval

Prevalence, odds ratios and 95% CIs are weighted by MEPS's sampling weights.

Referent group is "Unsatisfied"

* **ORs are adjusted for: 1) Individual-level factors**: Age; Gender; Race/ethnicity; Geographic region of residence; Education; Speake English at home; Marital status; Employment status; Year of MEPS survey; Poverty status; Ever delay, forego or make change in prescription medicine because of cost; Ever delay, forego or make change in treatment because of cost; Type of antidepressant used; severity of depression symptoms (PHQ2≥3); Depression; Bipolar disorder; Anxiety; Psychotherapy; Poor physical health; Poor mental health; Have cognitive limitations; Hypertension; Coronary heart disease; Angina; Myocardial infarction; Other heart diseases; Stroke; Diabetes; Arthritis; Asthma; Chronic bronchitis; Cancer. **2) Provider-level factors**: Provider specialty; Provider-patient gender concordance; Provider-patient race/ethnicity concordance. **3) Healthcare system factors**: Uninsured; Have usual source of payment; Payment source; Out-of-pocket payments; Counts of office visits.

their healthcare. The associations between satisfaction and adherence to antidepressant medications are reported in Table 2. In multivariable logistic regression analysis, patients who were satisfied with their healthcare were 26% (OR = 1.26, 95%CI: 1.08, 1.47) more likely to adhere to antidepressant medications, compared to those who were less satisfied, in the overall antidepressant user population. Furthermore, satisfied patients were less likely to completely discontinue antidepressant therapy during follow-up as compared to unsatisfied patients, OR = 0.83, 95%CI: 0.71, 0.96. Among a subpopulation of new users of antidepressants who had not yet potentially formed opinions about their provider with respect to antidepressant treatment outcomes, patients who were satisfied with their healthcare were also more likely to adhere with antidepressant as compared to those not satisfied, OR = 1.41, 95%CI: 1.06, 1.88. This finding among new antidepressant users shows that the observed satisfaction-adherence association could not have been due to potential reverse association.

## The roles of patient-provider communication and shared decision-making in the association between patient satisfaction and adherence to ADs

Fig 3 is a visual representation of the SEM that was used to assess the roles of patient-provider communication and shared decision-making in the association between patient satisfaction and adherence to ADs.

The individual items that make up the patient-clinician communication and SDM constructs are shown in Fig 3. Based on multiple model fit indices, the hypothesized SEM model was a good fit for the observed data: RMSEA = 0.01, TLI = 0.99, CFI = 0.99. Both constructs of patient-provider communication (β = 2.03, P-value<0.001) and SDM (β = 1.14, P-value<0.001) were strongly positively associated with satisfaction. This confirmed our hypothesis that satisfaction is a manifestation of patient-provider communication and SDM. However, the healthcare satisfaction-AD adherence association observed in the multivariable regression analysis attenuated (β = 0.01, P-value = 0.03) in the SEM framework in which we adjusted for the potential confounding effects of race/ethnicity, healthcare access (number of office visits, number of providers) and cognitive limitation. Only SDM was significantly associated with antidepressant adherence (β = 0.05, P-value = 0.01) in the specified model that was adjusted for race/ethnicity, healthcare access and cognitive limitation.

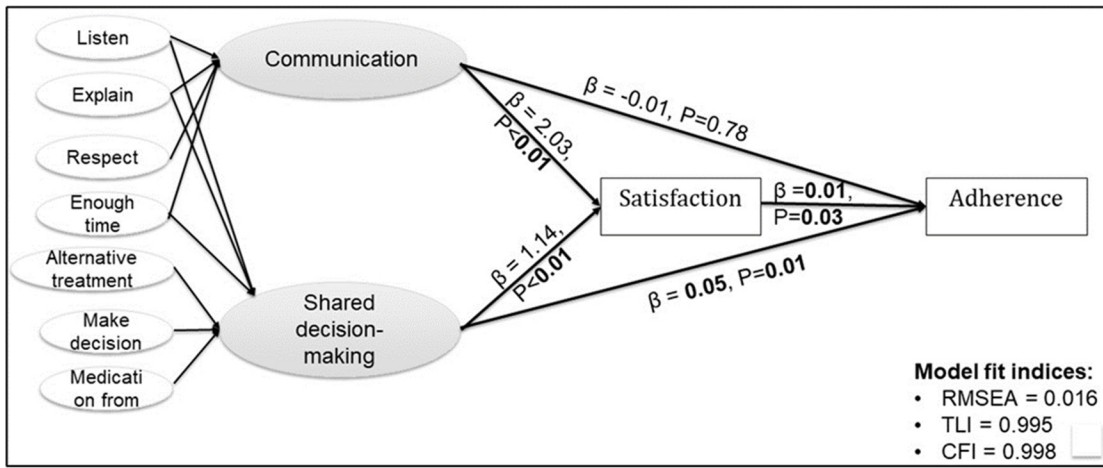

**Fig 3. A structural equation model.** The role of patient-provider communication and shared decision-making on patient satisfaction with care and adherence to antidepressants are depicted in this figure. The patient-clinician communication construct was created from participant responses (never, sometimes, usually, always) to the questions: how often the care provider (1) listen carefully to the patient. (2) explain to the patient. (3) show respect to the patient. (4) spend enough time with the patient? The SDM construct was defined from the four CAHPS items described above plus three additional questions about patients' satisfaction with their usual source of care provider: Does the usual source of care provider (1) usually ask about and show respect for medical, traditional, and alternative treatments that the person is happy with (never/sometimes/usually/always)?. (2) ask the person to help make decisions between a choice of treatments (never/sometimes/usually/always). (3) usually ask about prescription medications and treatments other doctors may give them (yes/no)?

## Discussion

Patient satisfaction with healthcare service was a strong predictor of antidepressant adherence among participants of MEPS who had reported the use of antidepressant and were considered to be depressed based on self-reported diagnosis. Specifically, the more satisfied patients were with the healthcare services that they received, the more likely they were to become adherent with antidepressant therapy. Our analysis revealed that satisfaction was a manifestation of the quality of patient-provider relationships with respect to communication and SDM. These findings suggested that patients who felt more engaged in making shared decisions about their care were not only more likely to be satisfied with the healthcare services that they had received but to also adhere to antidepressant therapy optimally. If confirmed in interventional studies, providers could potentially simultaneously improve depression patients' satisfaction with healthcare services as well as adherence to antidepressant therapy among depression patients.

Few studies have specifically applied a longitudinal study design to examine the association between patient satisfaction and adherence to ADs. Nonetheless, our findings are consistent with prior cross-sectional studies that have reported positive associations between patient satisfaction and antidepressant adherence [46, 47]. Our findings are also consistent with those of a systematic review of the associations between patient satisfaction and adherence to medications in general [48]. This review reported that higher patient satisfaction was significantly associated with better medication adherence, defined as compliance or persistence, among 16 studies that were included in the review.

Our results have empirically confirmed the general theory that patients are more likely to fully participate in making clinical decisions and to fully execute a recommended treatment regimen than if they were more satisfied. Rossom et. al. (2016) reported that depression patients were more likely to report being satisfied with care when providers engaged them in SDM (by soliciting patient preferences for care, providing treatment plans, etc) and communicated effectively (by asking questions and showing concerns, asking about suicide risks, etc)

[20]. This evidence is consistent with our findings of positive associations between patient satisfaction and both SDM and communication. It is plausible that participants with depression were more likely to adhere to antidepressant therapy because they trusted providers who communicated effectively and engaged them in making decisions about their depression treatment plans.

Overall, the results from this study provide insights that can be leveraged by providers to simultaneously improve both patient satisfaction and antidepressant adherence. Our findings imply that patient satisfaction rating is a strong predictor of adherence to antidepressant therapy among depression patients. Thus, providers could leverage patient satisfaction data that is routinely collected as part of reimbursement and quality improvement purposes to predict whether patients are going to become nonadherent to antidepressant therapy. Based on our explanatory analysis of the roles of patient-provider relationships, providers could potentially improve patient satisfaction as well as antidepressant adherence by improving patient-provider communication and greater involvement of patients in SDM.

## Limitations

Our study feature limitations that should be considered when interpreting the findings reported in this manuscript. First, we are unable to infer causal relationship between patient satisfaction and antidepressant adherence because our analysis involved observational data. Second, we may have misclassified patients as nonadherent even if they intentionally discontinued antidepressant therapy at the behest of a clinician because neither measure of MRA nor discontinuation are adequate to discern whether patients discontinued antidepressant therapy intentionally or not. However, such a bias may also have non-differential effects on the observed associations between patient satisfaction and adherence given that there is no plausible reason why one group of patients (e.g. satisfied) would be more likely to intentionally discontinue antidepressant therapy than the other (e.g. unsatisfied). Third, because participants were not directed to rate their providers based on depression care alone, the use of a global measure of patient satisfaction may not truly reflect the level of satisfaction with depression care. Fourth, while MRA is a validated measure of refill adherence, it was measured based on self-reported medication use and may therefore be liable to recall bias. Any such bias, however, would have had a differential effect on measured associations since we expect recall bias to be similar between the groups compared by levels of communication and SDM.

## Strengths

Our study features several strengths that make our findings robust. First, the longitudinal study design approach applied enabled us to delineate the temporal relationships between patient satisfaction (the exposure) and adherence to antidepressant (outcome). Thus, our findings provide data on how patient satisfaction levels may affect future adherence behaviors. Second, we applied sensitivity analysis to show that our findings were robust against the potential impact of reverse associations that tend to beset many observational studies. Third, by using both measures of refill adherence (MRA) and complete discontinuation of ADs, we showed that patient satisfaction was associated with both measures of antidepressant adherence thus further validating the robustness of our findings. Fourth, to the best of our knowledge, our study involved the largest sample to date that has been used to investigate the association between patient satisfaction and antidepressant adherence among a nationally representative population of depression patients who were treated with ADs. Fifth, unlike previous studies on this topic, ours elucidated the roles of patient-provider communication and SDM on the association between patient satisfaction and antidepressant adherence. Thus, our findings are

more informative with respect to how providers could potentially improve patient satisfaction and antidepressant adherence among antidepressant users.

## Conclusion

Patient satisfaction is a predictor of antidepressant adherence in this nationally representative sample of depression patients who were treated with ADs. Our findings suggest that patient satisfaction data is clinically valuable information, beyond meeting mandatory reimbursement requirements, and should therefore be leveraged by providers to predict treatment outcomes such as adherence to ADs.

## Author Contributions

**Conceptualization:** Macarius M. Donneyong.

**Data curation:** Macarius M. Donneyong.

**Formal analysis:** Macarius M. Donneyong.

**Methodology:** Macarius M. Donneyong.

**Writing – original draft:** Macarius M. Donneyong.

**Writing – review & editing:** Macarius M. Donneyong, Mary Bynum, Ameena Kemavor, Norah L. Crossnohere, Anne Schuster, John Bridges.

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
