## [Decision Letter · Decision Letter 0]

5 Sep 2023

PONE-D-23-11772PATIENT SATISFACTION WITH THE QUALITY OF CARE RECEIVED IS ASSOCIATED WITH ADHERENCE TO ANTIDEPRESSANT MEDICATIONS.PLOS ONE

Dear Dr. Donneyong,

Thank you for submitting your manuscript to PLOS ONE. After careful consideration, we feel that it has merit but does not fully meet PLOS ONE’s publication criteria as it currently stands. Therefore, we invite you to submit a revised version of the manuscript that addresses the points raised during the review process.

We look forward to receiving your revised manuscript.

Kind regards,

Qin Xiang Ng, MBBS, GDMH, MPH

Academic Editor

PLOS ONE

Reviewers' comments:

Reviewer's Responses to Questions

**Comments to the Author**

1. Is the manuscript technically sound, and do the data support the conclusions?

Reviewer #1: Partly

Reviewer #2: Partly

2. Has the statistical analysis been performed appropriately and rigorously? 

Reviewer #1: I Don't Know

Reviewer #2: No

3. Have the authors made all data underlying the findings in their manuscript fully available?

Reviewer #1: Yes

Reviewer #2: No

4. Is the manuscript presented in an intelligible fashion and written in standard English?

Reviewer #1: Yes

Reviewer #2: Yes

5. Review Comments to the Author

Reviewer #1: Dear colleagues,

Thank you for the opportunity to review this manuscript. The study examines the relationship between patient satisfaction, patient provider communication, including shared decision-making (SDM), and adherence to antidepressants. The manuscript has potential to contribute to the field and to provide additional evidence in support of the positive impact of SDM on patient outcomes. The manuscript has several strengths, including large sample and exploration of mechanisms linking patient satisfaction with antidepressant use. I have some suggestions to improve the manuscript, especially to help increase clarity and impact. Please see below.

Introduction

1. The introduction needs more development and citations. Many claims were made without proper citation. Please define SDM and cite appropriately. Other studies have discussed SDM in the context of depression. It would be helpful to describe some of these studies, even with just two sentences.

Methods

1. The DAG helps with visual representation but is not onto itself a conceptual framework. Was there a conceptual framework that guided this study. There is a lot of work that has been done in the area of patient-provider communication and SDM that could be helpful here.

2. Need to specify the study design in the methods section. Again, illustration is not enough to describe the study design.

3. The assessment section is unclear and underdeveloped. This section mentioned that patient consent was obtained. It seems contradictory to what was said earlier. Please clarify.

4. Also clarify why long-term adherence was selected for this study. Would the results be different if a shorter time frame was used. For participants who discontinued medication, were they included in the non-adherence pool?

5. There is insufficient data as to how SDM and communication were defined and measured in this study. Describe them and provide information on how they were scored, cut off points, and previous used of similar methods. Also include them as limitation section since no validated SDM scales were used and not for specific visits related to anti-depressant medication management. More information about the measurements is needed in general.

6. There were lots of various patient characteristics collected at baseline. However, they were not included/discussed in the results and discussion sections.

Results and Discussion

1. Exploration of the reverse association between antidepressant and satisfaction is a strength. However, there are some limitations. It is unclear if these are patients who have had previous hx of and treatment for depression (outside of the study timeframe). These are also not new patients.

2. No discussion about race, age, sex, and morbidity, including other mental health diagnoses, which have been shown to impact both patient satisfaction and participation in SDM. This is a major weakness and a missed opportunity.

3. Discussions and limitations are limited and could be further developed. They are also not well grounded in the literature.

Reviewer #2: Dear Editor,

Dear Authors,

Thank You for the opportunity to review statistical part of the manuscript entitled ‘PATIENT SATISFACTION WITH THE QUALITY OF CARE RECEIVED IS ASSOCIATED WITH ADHERENCE TO ANTIDEPRESSANT MEDICATIONS’.

The submitted manuscript addresses an important topic being in the broader area of social determinants of health. Although the impact of patient-provider communication and shared decision making on the patient's satisfaction and adherence to recommended treatment has been investigated for several years, the manuscript provides an interesting insights in the process.

Find below, please, some comments You may find useful in improving the scientific soundness of the manuscript:

Abstract

*there is no setting (country) mentioned, add, please;

*there are weighted counts presented, but there is no information on how it was weighted (country, population structure for the year) [the same comment is for the 1st paragraph under Results (page 80];

*under conclusions: the statement "patient satisfaction is a predictor ..." 'is' here is too definite, especially considering different factors which are associated with the adherence to AD recommended treatment which were not taken into account in the analyses, and the fact that after adjustment the SEM model results were not statistically significant, therefore showed that there is no prove for the association (the likelihood of mistake if concluded is too high). The same comment is for the title;

*conclusions cont.: moreover the authors state 'improving ...', the study however, did not evaluate the impact of intervention, therefore it is hard to conclude on what will happen after the mentioned improvement;

Introduction

*explain, please, the meaning of shortcuts at the first time of use (MDD, AD);

*the authors are referring to 'suboptimal adherence' as associated with relapse. Clarify please, what is the suboptimal. This is especially important considering the optimal adherence is hardly to achieve in majority of cases;

*patient-provider communication and patient involvement in shared decision-making are called as tenets. The observations and studies showed that these are the elements of the whole process, so clarify please, why they are called as tenets?

Methods

*data source, setting: there is no information about response rate / participation rate - add it please;

*the MEPS ran from 2010-2017, but only 2 calendar years 2015-2016; why not all or the last two? explain please. Considering the results of adjusted SEM were not statistically significant run the analyses on the whole sample and provide results at least in supplementary materials;

*it is not clearly provided what is 'the satisfaction'. Is that a satisfaction with the last visit? with the individual experience from the last year contacts with GP or each health care provider? or maybe it is satisfaction with overall health care performance? ... It should be provided what question(s) were asked to get the information about the satisfaction level.

*there is no information on how the depression / other mental health diagnoses have been identified, add it please;

*missing data: it is not clarified whether imputation was done as MCAR or MAR, and how it was determined whether missingness were at random or not

Statistical analysis

*SEM: it is not clear to me why only a limited number of factors were considered in the model, and others which are presented in the Fig.1 have been omitted. The SEM technique has been created to allow for consideration the interplays of several factors in the process. I strongly recommend using the whole conceptual framework to assess the impact of satisfaction.

Results

*page 8-1st paragraph: it refers to 'participants [...] not previously reporting using antidepressants'. Under methods it is mentioned that information was verified at the pharmacies. State clearly, please, what was the data source of AD use by study participant which were next considered for statistical analysis (and are presented in the manuscript). If those, who reported AD used were only verified at the pharmacies there is a likelihood of recall bias which should be mentioned under limitations.

*page 8-2nd paragraph: 'higher depression' - why it is called so? I suggest using more severe depression ...

*I suggest 'presenting' results instead of 'reporting' results;

*tab.1: footnotes are mistaken /confusing ... how n* represent mean or %+ represent standard deviation? for 'average annual out-of-pocket payment' how overall is 4990? ... maybe it would be better to change overall into 'the sample size'?

*tab.1: is SDT the column presenting the p-values or test statistics?

*tab.1: why educational level - college and above is considered as a barrier?

*tab.1: what were the criteria for poor physical health and poor mental health?

*tab.1: provide, please, the value for poverty level;

*tab.1: explain, please, what was considered to be a delay for purchasing medicine and in seeking medical care;

*tab.2: there are several factors used for adjustment, but the psychotherapy has been omitted ... why?

Discussion

*page 10 - 3rd paragraph: My guess is that the first sentence refers to more satisfied patients but not to all patients, but it requires clarification ...

6. PLOS authors have the option to publish the peer review history of their article (what does this mean?). If published, this will include your full peer review and any attached files.

Reviewer #1: No

Reviewer #2: No

---

## [Author Response · Author response to Decision Letter 0]

25 Oct 2023

Reviewer #1:

Dear colleagues,

Thank you for the opportunity to review this manuscript. The study examines the relationship between patient satisfaction, patient provider communication, including shared decision-making (SDM), and adherence to antidepressants. The manuscript has potential to contribute to the field and to provide additional evidence in support of the positive impact of SDM on patient outcomes. The manuscript has several strengths, including large sample and exploration of mechanisms linking patient satisfaction with antidepressant use. I have some suggestions to improve the manuscript, especially to help increase clarity and impact. Please see below.

Introduction

Reviewer comment 1: The introduction needs more development and citations. Many claims were made without proper citation. Please define SDM and cite appropriately. Other studies have discussed SDM in the context of depression. It would be helpful to describe some of these studies, even with just two sentences.

Author response: Thanks for your comment. We have edited it according to your suggestion, see introduction section (page #3).

Methods

Reviewer comment 1: The DAG helps with visual representation but is not onto itself a conceptual framework. Was there a conceptual framework that guided this study? There is a lot of work that has been done in the area of patient-provider communication and SDM that could be helpful here.

Author response: It appears our understanding of a conceptual framework differs from that of the reviewer, as such, we would like to clarify and justify our use of a DAG in this context. We not only visually represent the inter-relationships between variables and the expected outcomes, but we also provided an explanation of the theoretical basis through which the measured covariates, exposures and mediators may interact to produce the effect of medication nonadherence. This set us is consistent with the standard definitions of a conceptual framework, here are a few examples of the definition of a conceptual framework: 

1. https://resources.nu.edu/c.php?g=1013602&p=7661246

2. https://dovetail.com/research/conceptual-framework/

3. https://www.scribbr.com/methodology/conceptual-framework/#:~:text=A%20conceptual%20framework%20is%20a,existing%20studies%20about%20your%20topic. 

Reviewer comment 2: Need to specify the study design in the methods section. Again, illustration is not enough to describe the study design.

Author response: Thanks for your suggestion, we have now expanded on the description of the study design used (pages: 4-5). 

Reviewer comment 3: The assessment section is unclear and underdeveloped. This section mentioned that patient consent was obtained. It seems contradictory to what was said earlier. Please clarify.

Author response: Thanks for your comment. We have now edited this section (pages: 5-6). 

Reviewer comment 4: Also clarify why long-term adherence was selected for this study. Would the results be different if a shorter time frame was used? For participants who discontinued medication, were they included in the non-adherence pool?

Author response: MEPS captured monthly prescription medication refill data for the entire calendar year and do not provide the specific refill dates, as such it is not possible to measure 6-month adherence.

Re: your comment about discontinuation, the MRA measure takes into account medication discontinuation. 

Reviewer comment 5: There is insufficient data as to how SDM and communication were defined and measured in this study. Describe them and provide information on how they were scored, cut-off points, and previous used of similar methods. Also include them as limitation section since no validated SDM scales were used and not for specific visits related to anti-depressant medication management. More information about the measurements is needed in general.

Author response: We defined both SDM and communication from CAHPS using standard definitions and have duly provided the references for these measures (page: 7)

Reviewer comment 6: There were lots of various patient characteristics collected at baseline. However, they were not included/discussed in the results and discussion sections.

Author response: Our covariate selection was informed by the DAG as we described in the manuscript. We are curious about the “various patient characteristics collected at baseline” you would like us to include and the justification for the inclusion of these variables. 

Results and Discussion

Reviewer comment 1: Exploration of the reverse association between antidepressant and satisfaction is a strength. However, there are some limitations. It is unclear if these are patients who have had previous hx of and treatment for depression (outside of the study timeframe). These are also not new patients.

Author response: Thanks for your comment. Yes, they are not new patients. We have clearly mentioned it in the study design section. Just like all observational databases, it is not possible to conclusively determine if an individual had never being diagnosed with depression or had never used an antidepressant. The MEPS data is left truncated, again, like all observational databases, as such information regarding participants prior to their participation in the survey is unknown. That being said, we considered patients who had no records of antidepressant use for the entire calendar period in Year 1 (baseline) but were observed to be using antidepressants in Year 2, were considered as “new users”. This operational definition of “new users” is a standard approach in the field of pharmacoepidemiology. 

Reviewer comment 2: No discussion about race, age, sex, and morbidity, including other mental health diagnoses, which have been shown to impact both patient satisfaction and participation in SDM. This is a major weakness and a missed opportunity.

Author response: We would appreciate an elaboration on this comment. We included race, age, sex, and morbidity, including other mental health diagnoses as covariates in the analysis. As such, the associations measured between patient satisfaction and antidepressant adherence are independent of these covariates; our goal was to evaluate whether patient satisfaction is an independent predictor of adherence to antidepressant.

Reviewer comment 3: Discussions and limitations are limited and could be further developed. They are also not well grounded in the literature.

Author response: We would appreciate it if you would kindly be specific about this comment (page 12). 

Reviewer #2:

Dear Editor,

Dear Authors, Thank You for the opportunity to review statistical part of the manuscript entitled ‘PATIENT SATISFACTION WITH THE QUALITY OF CARE RECEIVED IS ASSOCIATED WITH ADHERENCE TO ANTIDEPRESSANT MEDICATIONS’.

The submitted manuscript addresses an important topic being in the broader area of social determinants of health. Although the impact of patient-provider communication and shared decision-making on the patient's satisfaction and adherence to recommended treatment has been investigated for several years, the manuscript provides an interesting insight in the process.

Find below, please, some comments You may find useful in improving the scientific soundness of the manuscript:

Abstract

We appreciate the critiques and suggestions by the reviewer and have specifically included information regarding the setting for this study. 

Reviewer comment 1: weighted counts presented, but there is no information on how it was weighted (country, population structure for the year): 

Author response: The weights applied in this study are the MEPS sampling survey weights, these survey weights reflect adjustments for survey nonresponse and adjustments to population control totals. These survey weights are different from what the reviewer inferred in the comment.

Reviewer comment 2: under conclusions: the statement "patient satisfaction is a predictor ..." 'is' here is too definite, especially considering different factors which are associated with the adherence to AD recommended treatment which were not taken into account in the analyses, and the fact that after adjustment the SEM model results were not statistically significant, therefore showed that there is no prove for the association (the likelihood of mistake if concluded is too high). The same comment is for the title;

Author response: We have modified the conclusion to address this concern. 

Introduction

Reviewer comment 1: *explain, please, the meaning of shortcuts at the first time of use (MDD, AD);

Author response: Thanks for your suggestion. We have now corrected it (page 3). 

Reviewer comment 2: *the authors are referring to 'suboptimal adherence' as associated with relapse. Clarify please, what is the suboptimal. This is especially important considering the optimal adherence is hardly to achieve in majority of cases;

Author response: The goal of antidepressant therapy, and pharmacotherapy in general, is to attain optimal adherence. There are various thresholds for optimal adherence and none of them is totalitarian, i.e. 100%; instead, optimal adherence as defined based on the common measures of adherence, e.g. proportion of days covered (PDC) and medication refill adherence (MRA), are based on >80% thresholds. 

Reviewer comment 3: *patient-provider communication and patient involvement in shared decision-making are called as tenets. The observations and studies showed that these are the elements of the whole process, so clarify please, why they are called as tenets?

Author response: Patient-provider communication and patient involvement in shared decision-making are foundational to the patient-provider relationship, as such it is appropriate to refer to them as tenets. We appreciate the semantic differences. 

Methods

Reviewer comment 1: *data source, setting: there is no information about response rate / participation rate - add it please;

Author comment: Thanks for pointing this out, we have now provided this information (page 4).

Reviewer comment 2: *the MEPS ran from 2010-2017, but only 2 calendar years 2015-2016; why not all or the last two? explain please. Considering the results of adjusted SEM were not statistically significant run the analyses on the whole sample and provide results at least in supplementary materials;

Author response: Our analysis used all the data from 2010 - 2017. As we mentioned in the manuscript, the 2015-2016 period was used only for illustrative purposes to elucidate how we constructed the study cohorts (page 4). 

Reviewer comment 3: *it is not clearly provided what is 'the satisfaction'. Is that a satisfaction with the last visit? with the individual experience from the last year contacts with GP or each health care provider? or maybe it is satisfaction with overall health care performance? ... It should be provided what question(s) were asked to get the information about the satisfaction level.

Author response: We have now clarified which time point patient satisfaction was assessed. We already included in the manuscript the question that was asked to elicit patient ratings of their satisfaction with care: “MEPS participants who reported having a usual source of care (USC) were asked to rate the healthcare that they had received from all doctors and other health providers, from 0 (worst health care possible) to 10 (best health care possible).” 

Reviewer comment 4: *there is no information on how the depression / other mental health diagnoses have been identified, add it please;

Author response: Thanks for your comment, we have now included information to address this concern (page 5).

Reviewer comment 5: *missing data: it is not clarified whether imputation was done as MCAR or MAR, and how it was determined whether missingness were at random or not

Author response: We have now clarified that we imputed based on the assumption of missing at random (page 7).

Reviewer comment 6: Statistical analysis: *SEM: it is not clear to me why only a limited number of factors were considered in the model, and others which are presented in the Fig.1 have been omitted. The SEM technique has been created to allow for consideration the interplays of several factors in the process. I strongly recommend using the whole conceptual framework to assess the impact of satisfaction.

Author response: We appreciate this comment and have taken the opportunity to clarify that our SEM analysis indeed adjusted for the baseline covariates; we did not show all the covariates in the SEM figure for visual clarity. 

Results

Reviewer comment 1: *page 8-1st paragraph: it refers to 'participants [...] not previously reporting using antidepressants'. Under methods, it is mentioned that information was verified at the pharmacies. State clearly, please, what was the data source of AD use by study participant which were next considered for statistical analysis (and are presented in the manuscript). If those, who reported AD used were only verified at the pharmacies there is a likelihood of recall bias which should be mentioned under limitations.

Author response: I we understand correctly, your concern is about potential recall bias about self-reported medication use, correct? We appreciate the concern and have now added as a limitation the potential impact of recall bias in self-reported medication use. 

Reviewer comment 2: *page 8-2nd paragraph: 'higher depression' - why it is called so? I suggest using more severe depression ...

Author response: Thanks for your suggestion, we have now replaced the word “higher” with “severe”.

Reviewer comment 3: *I suggest 'presenting' results instead of 'reporting' results;

Author response: We appreciate the semantic suggestion, but we believe it is appropriate to report results.

Reviewer comment 4: *tab.1: footnotes are mistaken /confusing ... how n* represent mean or %+ represent standard deviation? for 'average annual out-of-pocket payment' how overall is 4990? ... maybe it would be better to change overall into 'the sample size'?

Author response: Thanks for your suggestion, we have made the appropriate corrections.

Reviewer comment 5: *tab.1: is SDT the column presenting the p-values or test statistics?

Author response: The SDT represents standardized difference test which is a measure of the effect size between the satisfied and unsatisfied patient groups. SDTs are independent of sample size and multiple hypothesis testing effects that are known limitations of P-values. 

Reviewer comment 6: *tab.1: why educational level - college and above is considered as a barrier?

Author response: We have no changed the subtitle from “barriers” to “factors” in Table 1.

Reviewer comment 7: *tab.1: what were the criteria for poor physical health and poor mental health?

Author response: Thanks for your comment. Collected on the Self-Administered Questionnaire and, the Preventive Self-Administered Questionnaire, the MEPS-HC includes two validated adult mental health scales. The Kessler Psychological Distress Scale (K6) and the two-item Patient Health Questionnaire depression screener (PHQ-2) are asked twice per panel, during interview rounds 2 and 4.

Reviewer comment 8: *tab.1: provide, please, the value for poverty level;

Author response: Thanks for your comment. “The MEPS administers a detailed income supplement that produces income and poverty status estimates that are post-stratified to match the Current Population Survey’s distributions, the nation’s official source of poverty statistics. Depending on the size of the reporting unit and the age of the household head, respondents were shown a card that provided family income ranges corresponding to five poverty status categories. The five categories included: 1. below poverty; 2. 100 to 150 percent of poverty; 3. 150 to 200 percent of poverty; 4. 200 to 300 percent of poverty; and 5. 300 percent or more.”

Reviewer comment 9: *tab.1: explain, please, what was considered to be a delay for purchasing medicine and in seeking medical care;

Author response: We have now clarified that the delays in purchasing medicine and seeking medical care were due to cost.

Reviewer comment 10: *tab.2: there are several factors used for adjustment, but the psychotherapy has been omitted ... why?

Author response: We appreciate this comment. We indeed adjusted for psychotherapy in our analysis but inadvertently left it out in the footnote of Table 2. 

Discussion

Reviewer comment 1: *page 10 - 3rd paragraph: My guess is that the first sentence refers to more satisfied patients but not to all patients, but it requires clarification ...

Author response: Thank you very much for pointing out this error. You are indeed correct that it should be “…more satisfied…..”. We have now fixed the error.

---

## [Decision Letter · Decision Letter 1]

20 Nov 2023

PONE-D-23-11772R1PATIENT SATISFACTION WITH THE QUALITY OF CARE RECEIVED IS ASSOCIATED WITH ADHERENCE TO ANTIDEPRESSANT MEDICATIONS.PLOS ONE

Dear Dr. Donneyong,

Thank you for submitting your manuscript to PLOS ONE. After careful consideration, we feel that it has merit but does not fully meet PLOS ONE’s publication criteria as it currently stands. Therefore, we invite you to submit a revised version of the manuscript that addresses the points raised during the review process.

We look forward to receiving your revised manuscript.

Kind regards,

Qin Xiang Ng, MBBS, GDMH, MPH

Academic Editor

PLOS ONE

Journal Requirements:

Reviewers' comments:

Reviewer's Responses to Questions

**Comments to the Author**

1. If the authors have adequately addressed your comments raised in a previous round of review and you feel that this manuscript is now acceptable for publication, you may indicate that here to bypass the “Comments to the Author” section, enter your conflict of interest statement in the “Confidential to Editor” section, and submit your "Accept" recommendation.

Reviewer #2: (No Response)

2. Is the manuscript technically sound, and do the data support the conclusions?

Reviewer #2: Yes

3. Has the statistical analysis been performed appropriately and rigorously? 

Reviewer #2: Yes

4. Have the authors made all data underlying the findings in their manuscript fully available?

Reviewer #2: Yes

5. Is the manuscript presented in an intelligible fashion and written in standard English?

Reviewer #2: Yes

6. Review Comments to the Author

Reviewer #2: Dear Authors,

The submitted revision addresses majority of issues mentioned in my primary review. There are only two elements which should be improved, in my opinion. These are:

1) The Kessler Psychological Distress Scale (K6) and the two-item Patient Health Questionnaire depression screener (PHQ-2) used to assess physical and mental health. The explanation was provided in the 'Authors' response', this information, however, is still not available to the readers. I suggest adding it under Methods.

2) The similar comment is for data collection strategy used for poverty level. Add, please, that it was 'as declared by the respondent'

Reviewer

7. PLOS authors have the option to publish the peer review history of their article (what does this mean?). If published, this will include your full peer review and any attached files.

Reviewer #2: No

---

## [Author Response · Author response to Decision Letter 1]

26 Nov 2023

Thank you for the second opportunity to further revise our manuscript. We are most grateful to the reviewers for their very constructive additional comments. We have fully addressed the comments below: 

Reviewer #2: Dear Authors, The submitted revision addresses majority of issues mentioned in my primary review. There are only two elements which should be improved, in my opinion. These are:

1) The Kessler Psychological Distress Scale (K6) and the two-item Patient Health Questionnaire depression screener (PHQ-2) used to assess physical and mental health. The explanation was provided in the 'Authors' response', this information, however, is still not available to the readers. I suggest adding it under Methods.

Response: We agree that adding this information to the manuscript has improved the clarity for how these variables were measured. We have added a description for the measurement of these variables on page #7.

2) The similar comment is for data collection strategy used for poverty level. Add, please, that it was 'as declared by the respondent'

Response: Similar to our response above, we have added a description for the measurement poverty status on page #7.

---

## [Decision Letter · Decision Letter 2]

6 Dec 2023

PATIENT SATISFACTION WITH THE QUALITY OF CARE RECEIVED IS ASSOCIATED WITH ADHERENCE TO ANTIDEPRESSANT MEDICATIONS.

PONE-D-23-11772R2

Dear Dr. Donneyong,

We’re pleased to inform you that your manuscript has been judged scientifically suitable for publication and will be formally accepted for publication once it meets all outstanding technical requirements.

Kind regards,

Qin Xiang Ng, MBBS, GDMH, MPH

Academic Editor

PLOS ONE

Additional Editor Comments (optional):

Reviewers' comments:

Reviewer's Responses to Questions

**Comments to the Author**

1. If the authors have adequately addressed your comments raised in a previous round of review and you feel that this manuscript is now acceptable for publication, you may indicate that here to bypass the “Comments to the Author” section, enter your conflict of interest statement in the “Confidential to Editor” section, and submit your "Accept" recommendation.

Reviewer #2: All comments have been addressed

2. Is the manuscript technically sound, and do the data support the conclusions?

Reviewer #2: Yes

3. Has the statistical analysis been performed appropriately and rigorously? 

Reviewer #2: Yes

4. Have the authors made all data underlying the findings in their manuscript fully available?

Reviewer #2: Yes

5. Is the manuscript presented in an intelligible fashion and written in standard English?

Reviewer #2: Yes

6. Review Comments to the Author

Reviewer #2: Authors addressed properly all the issues. In my opin opinion the manuscript fits the criteria to be published.

7. PLOS authors have the option to publish the peer review history of their article (what does this mean?). If published, this will include your full peer review and any attached files.

Reviewer #2: **Yes: **Aleksander Galas

---

## [Editor Report · Acceptance letter]

22 Dec 2023

PONE-D-23-11772R2 

PLOS ONE

Dear Dr. Donneyong, 

I'm pleased to inform you that your manuscript has been deemed suitable for publication in PLOS ONE. Congratulations! Your manuscript is now being handed over to our production team.

Kind regards, 

on behalf of

Dr. Qin Xiang Ng 

Academic Editor

PLOS ONE